

# Trends in bacterial and fungal communities in ant nests observed with Terminal-Restriction Fragment Length Polymorphism (T-RFLP) and Next Generation Sequencing (NGS) techniques—validity and compatibility in ecological studies

Stafva Lindström[1,2,3], Owen Rowe[4,5,6], Sari Timonen[6], Liselotte Sundström[1,2,3] and Helena Johansson[1,2]

[1] Centre of Excellence in Biological Interactions, Department of Biosciences, University of Helsinki, Helsinki, Finland
[2] Tvärminne Zoological Station, University of Helsinki, Hanko, Finland
[3] Organismal and Evolutionary Biology, University of Helsinki, Helsinki, Finland
[4] Umeå Marine Sciences Centre, Umeå University, Hörnefors, Sweden
[5] Department of Ecology and Environmental Science, Umeå University, Umeå, Sweden
[6] Department of Microbiology, University of Helsinki, Helsinki, Finland

Corresponding author
Stafva Lindström,
stafva.lindstrom@helsinki.fi

## ABSTRACT

Microbes are ubiquitous and often occur in functionally and taxonomically complex communities. Unveiling these community dynamics is one of the main challenges of microbial research. Combining a robust, cost effective and widely used method such as Terminal Restriction Fragment Length Polymorphism (T-RFLP) with a Next Generation Sequencing (NGS) method (Illumina MiSeq), offers a solid alternative for comprehensive assessment of microbial communities. Here, these two methods were combined in a study of complex bacterial and fungal communities in the nest mounds of the ant *Formica exsecta*, with the aim to assess the degree to which these methods can be used to complement each other. The results show that these methodologies capture similar spatiotemporal variations, as well as corresponding functional and taxonomical detail, of the microbial communities in a challenging medium consisting of soil, decomposing plant litter and an insect inhabitant. Both methods are suitable for the analysis of complex environmental microbial communities, but when combined, they complement each other well and can provide even more robust results. T-RFLP can be trusted to show similar general community patterns as Illumina MiSeq and remains a good option if resources for NGS methods are lacking.

## INTRODUCTION

Microbes are the most numerous organisms on earth (*Bertrand et al., 2011*). Their small size and the difficulty of cultivating many (if not most) species has spurred the development of molecular techniques to study the taxonomy, community structure and functions of natural microbial communities. Several techniques such as Automated Ribosomal Intergenic Spacer Analysis (ARISA) (*Ramette, 2009*), Temperature Gradient Gel Electrophoresis (TGGE), Denaturing Gradient Gel Electrophoresis (DGGE) (*Nocker, Burr & Camper, 2007*) and Terminal Restriction Fragment Length Polymorphism (T-RFLP) (*Liu et al., 1997*; *Kent et al., 2003*; *Cao et al., 2013*) have been developed for the assessment of complex microbial communities. The semi-quantitative properties (*Blackwood et al., 2003*; *Aiken, 2011*), high reproducibility, low cost (*Thies, 2007*) and the technical and analytical straightforwardness of T-RFLP has cemented it as one of the leading fingerprinting methods for several decades (*Schütte et al., 2008*). T-RFLP has successfully been used to evaluate the effects of spatial and temporal variation in the structure of microbial communities (*Schütte et al., 2008*; *Robinson et al., 2009*; *Barnes et al., 2016*). However, to obtain taxonomic identity of microbial community members, T-RFLP is usually combined with time-consuming and expensive preparation and sequencing of clone libraries (*Van Elsas & Boersma, 2011*).

Recent advances in high-throughput DNA sequencing techniques have made NGS (Next Generation Sequencing) methods such as Illumina MiSeq/HiSeq, IonTorrent and 454 pyrosequencing attractive alternatives for analysis of microbial communities (*Teeling & Glöckner, 2012*). In contrast to T-RFLP, sequence data generated by these methods reveal taxonomic identity to the extent that designated sequence data is available (*Weissbrodt et al., 2012*; *De La Fuente et al., 2014*; *Prakash et al., 2014*). However, NGS datasets can be extensive, require more complex bioinformatic interpretation and the protocols and methods used for microbial NGS analysis are far from standardized (*Amend, Seifert & Bruns, 2010*; *Tedersoo et al., 2015*; *Balint et al., 2016*). Complementing T-RFLP with NGS techniques can, however, offer a convenient way of generating and analyzing metagenomic microbial data, including the addition of taxonomic information achieved by NGS sequencing, to the T-RFLP fingerprints. This is especially useful for longitudinal studies, where substantial T-RFLP-fingerprinting data, including data generated prior to the NGS era, are available. Furthermore, different environments facilitate and promote microbiomes of different composition and diversity and the analysis should be carefully adapted to the specific study at hand (*Buttigieg & Ramette, 2014*; *Faust et al., 2015*; *Weiss et al., 2016*). The few studies to date on bacteria, indicate that the patterns and trends in data generated by the two techniques are largely congruent (rumen bacteria (*De La Fuente et al., 2014*), bacteria in Antarctic soil (*Van Dorst et al., 2014*), bacteria in tropical soil (*Supramaniam et al., 2016*)). Studies that utilize both T-RFLP and NGS data to explore fungal communities appear absent from the literature, as do studies comparing both bacterial and fungal communities using both techniques.

Ants are frequent in most terrestrial environments (*Vander Meer, 2012*), often being referred to as ecosystem engineers due to their skills in manipulating the abiotic and biotic properties of soil (*Dauber, Schroeter & Wolters, 2001*; *Jurgensen et al., 2008*). While

building their nests, the bioturbation and the selective picking of appropriate construction materials create a soil chemistry that differs significantly from the one of the background soil (*Kilpeläinen et al., 2007*; *Dostál et al., 2005*). The distinct nest environment is clearly reflected by the nest microbiome: the nests of several mound building ant species contain a substantially higher microbial biomass (*Dauber, Schroeter & Wolters, 2001*), more numerous and diverse fungal colonies (*Duff et al., 2016*) and significantly different microbial assemblages (*Boots et al., 2012*), than those of the reference soils.

The aim of this study was to evaluate the congruence of Illumina MiSeq and T-RFLP data, based on the bacterial 16s and fungal ITS areas of the rRNA genes from complex environmental communities. To do this, the microbial communities inside the nests of the mound building ant *Formica exsecta* (Nylander, 1846) were sampled over a two-year period. The bacterial and fungal communities in the *F. exsecta* mounds have not been studied previously, but here molecular microbial data of active nest mounds is reported, generated with both methodologies. The relative abundance patterns, diversities and sampling efficiencies generated by the two molecular techniques were compared, and the similarity of the functional, spatial, and temporal patterns were evaluated. The extent to which T-RFLP can be complemented with the taxonomic information gained from the NGS data was also assessed.

## MATERIALS & METHODS

### Ant species, study site and sampling

Ants of the genus *Formica* are common in Northern Eurasia (*Collingwood, 1979*; *Czechowski, Radchenko & Czhechowska, 2002*; *Goropashnaya et al., 2007*). Most species construct perennial above-ground mounds from litter and soil particles gathered from the surrounding forest floor (*Jurgensen et al., 2008*). The ant *Formica exsecta* used in this study is common (*Douwes et al., 2012*), and inhabits meadows, sunny woodland openings, and forest clearings (*Sundstrom, Chapuisat & Keller, 1996*). The study sites are located on islands in the south-western archipelago of Finland, (Fig. S1) near Tvärminne zoological station (59°84′196″N, 23°20′182″E), where populations of *F. exsecta* have been monitored and studied since 1994 (e.g., *Sundstrom, Chapuisat & Keller, 1996*; *Sundström, Keller & Chapuisat, 2003*; *Vitikainen, Haag-Liautard & Sundström, 2011*). The islands represent typical biotopes of the SW coast of Finland, where granite outcrops, pine spinneys, and dry meadows alternate with intermixed lusher patches of vegetation. The soil is classified as lepotosol (http://www.fao.org/soils-portal/soil-survey/soil-classification/world-reference-base/en/soil) of varying thickness due to the topographic factors. Scots pine and Norway spruce dominate, together with ericaceous shrubs and species of *Deschampsia* and *Festuca*. Sampling was approved by the land owner (Tvärminne Zoological station), and no endangered or protected species were included in this study. Sampling was carried out once a month in May, June, July, August and September in both 2013 and 2014. Four nest mounds were sampled, two per island (F12 and F120 on Furuskär, J30 and J40 on Joskär, (Fig. S1)) on each consecutive sampling occasion, totalling 40 samples. The samples (∼0.2 l) were collected by hand from the inside of the mound at the depth of 10–15 cm

using sterile gloves and placed in clean plastic bags. Samples were stored at −80 °C until extraction of DNA.

## DNA extraction and PCR amplification for T-RFLP

DNA was extracted from a ∼0.25 g subsample of nest mound material using the PowerSoil® DNA Isolation Kit (MoBio Laboratories Inc., Carlsbad, CA, USA) according to the manufacturer's instructions, with one exception being the use of TissueLyser II (Qiagen, Valencia, CA, USA) for 3 min at 20,000 rpm during the cell lysis phase. DNA was eluted in 100 µl of elution buffer, and the same DNA extractions were used for the T-RFLP analysis and Illumina MiSeq sequencing as described below.

Sequences from the fungal ribosomal ITS2 region were amplified with the TAMRA-tagged forward primer fITS7 (GTGA(A/G)TCATCGAATCTTTG, *Ihrmark et al., 2012*), and the reverse primer ITS4 (TCCTCCGCTTATTGATATGC, *White et al., 1990*). PCR reactions were performed in a final volume of 15 µl, containing 0.75 µl of each 10 µM primer, 3.0 µl GC 5 × buffer, 0.3 µl of 10 mM dNTP's, 0.15 µl of Phusion$^{TM}$ DNA polymerase (Thermo Fisher, Waltham, MA, USA), 0.75 µl of DNA template and 8.7 µl Milli-Q water. The PCR conditions were 30 s at 98 °C; followed by 34 cycles of 10 s at 98 °C, 30 s at 54 °C, and 30 s at 72 °C, followed by a final 5 min elongation step at 72 °C. The PCRs were performed in duplicates and pooled into one product. Sequences from the bacterial 16S rRNA region were amplified with the FAM-tagged forward primer 27F (AGAGTTTGATC(A/C)TGGCTCAG, *Weisburg et al., 1991*; *Chung et al., 2004*) and the reverse primer 1387R (GGGCGG(A/T)GTGTACAAGGC, *Wade et al., 1998*). PCR reactions were performed as described above, with the exception of the step in cycling before final elongation that was 40 s for bacteria.

Pooled duplicate PCR products from bacteria or fungi were purified with Agencourt AMPure XP beads (1 DNA:1.2 AMPure, Beckman Coulter) and digested with the restriction endonucleases HaeIII and MspI (Thermo Fisher, Waltham, MA, USA). The reaction volumes were 10 µl purified PCR product, 1 µl (10U) of each enzyme, 2 µl 10 × FastDigest buffer and 17 µl nuclease free water. Samples were digested for 5 min at 37 °C. The labelled terminal restriction fragments (T-RFs) were separated on an ABI-sequencer (model 3730; Applied Biosystems, Foster City, CA, USA) with the internal size standard GeneScan 500 ROX (Thermo Fisher, Waltham, MA, USA) and analysed using GeneMapper v.5 (Applied Biosystems, Foster City, CA, USA).

## Analysis of T-RFLP data

Peak data of T-RFs within the range of 70–400 bp and above a threshold of units of fluorescence of 100 for bacteria, and 70 for fungi were extracted from GeneMapper and imported into T-rex (http://trex.biohpc.org) for filtering of noise (removal of peaks not considered to comprise unique T-RFs), alignment and binning (*Culman et al., 2009*). Exact matches were used, i.e., only single base errors (±1 bp), and height of peaks (*Dickie & Fitzjohn, 2007*; *Aiken, 2011*; *Fredriksson, Hermansson & Wilén, 2014*) were used to determine the relative abundance. The peak data produced by the two enzymes (HaeIII and MspI) were filtered, aligned and binned in T-rex separately and pooled together (*De La*

*Fuente et al., 2014*). The only instances where the data from the two enzymes were analysed separately was for matching the experimental T-RFs with the virtual T-RFs (described below) and for assessing the level of correlation between the ranked relative abundances of T-RFs and OTUs.

### Illumina MiSeq library preparation and sequencing

The fungal ITS region was amplified with the primers fITS7 (GTGA(A/G)TCATCGAATCTTTG, *Ihrmark et al., 2012*) and ITS4 (TCCTCCGCT-TATTGATATGC, *White et al., 1990*) modified with partial Illumina TruSeq adaptor sequence. An initial PCR was performed with Phusion Hot Start II polymerase (Thermo Fischer) with a denaturation step of 98 °C for 30 s, followed by 15 cycles at 98 °C for 10 s, 65 °C for 30 s, 72 °C for 10 s followed by a final extension for 5 min at 72 °C. PCR samples were purified with Exonuclease I (Thermo Scientific, Waltham, MA, USA) and Thermosensitive Alkaline Phosphatase (FastAP; Thermo Scientific, Waltham, MA, USA). A second PCR was performed on the template products with full-length TruSeq P5 and index-containing P7 adapters. Cycling conditions were the same as the first library preparation PCR, but with an increased number of cycles (18). DNA libraries were quantified with Qubit (Invitrogen, Carlsbad, CA, USA) and library quality assessed with a Bioanalyzer 2100 (Agilent, Santa Clara, CA, USA), before pooling to equimolar amounts and submitted to sequencing. Due to technical requirements, the same reverse primer as for the T-RFLP could not be used. The bacterial 16s rRNA region was amplified using the primers 27F (AGAGTTTGATC(A/C)TGGCTCAG, *Weisburg et al., 1991*; *Chung et al., 2004*) and pD' (GTATTACCGCGGCTGCTG, *Edwards et al., 1989*). Libraries were otherwise constructed using the protocol described above. Sequencing was carried out using an Illumina MiSeq v2 600 cycle kit in paired-end mode at the DNA Sequencing and Genomics Laboratory, Institute of Biotechnology, University of Helsinki.

### Illumina MiSeq—bioinformatics data analysis

Read filtering and OTU clustering (at 97% identity) was performed using UPARSE v. 8.1 (*Edgar, 2013*; *Edgar & Flyvbjerg, 2015*), Table S1. Database SILVAv123 (*Quast et al., 2013*) was used as reference database for alignment of the bacterial sequences, and UNITE v7 (*Kõljalg et al., 2013*) for the fungal sequences. For taxonomic classification of the bacterial OTUs, the reference database RDP16s training set v.14 (*Wang et al., 2007*) was used, and for the fungal OTUs the RDP ITS Warcup training set v.4 (*Deshpande et al., 2016*) was used. The queries against the databases were done by using the RDP Naïve Bayesian Classifier with bootstrap cut-off at 80% (*Wang et al., 2007*). Singletons, doubletons, and sequences not identified to the level of kingdom were removed prior to further analysis. The unprocessed sequences are available at NCBI Sequence Read Archive, Bioproject number PRJNA399258.

### Species diversity, effective sampling estimates, and functional organisation

The two molecular methods were compared based on species accumulation curves, performed in the statistical framework R using the function specaccum in package vegan v.

2.4-5 (*Oksanen et al., 2011*), and Good's estimate (*Good, 1953*) of sampling coverage. The curves visualize the level of sampling effort as a function of the accumulated number of samples with a more saturated curve indicating a better sampling effort. Good's estimate indicates the proportion of the population that has been captured by the sampling or sequencing (*Good, 1953*). Furthermore, the number of T-RFs and OTUs were counted, and the mean value of the total fungal and bacterial diversity was estimated by calculating respective Shannon-Wiener's diversity index ($H'$) on the T-RFs and OTUs using the diversity function in R, package vegan (*Oksanen et al., 2011*).

The functional organization *(Fo)* of bacteria and fungi obtained with the two molecular methods, was compared by estimating Pareto-Lorenz curves (performed in JMP v.11, SAS Institute Inc.). To assess the degree of functional organization, the cumulative normalized number of T-RFs or OTUs were plotted against their respective cumulative normalized height (T-RFs) or sequence abundance (OTUs) (*Marzorati et al., 2008*). This always creates a convex curve as the values are positive by default. The deviation of the curve from the 45° diagonal line (representing perfect evenness with no functional organization) indicates the degree of functional organization (*Mertens, Boon & Verstraete, 2005*; *Wittebolle et al., 2009*). The *Fo* value is determined from the $y$-axis where the curve intercepts with the 20% $x$-axis line. A *Fo* value of 25% represents a community of high evenness with no distinct structure in terms of species dominance. A community at the *Fo* value of 45% is more functionally organized due to lower evenness, and a value of 80% for the *Fo* stands for a highly specialized community, dominated by a low number of species on which the functional stability depends. The level of *Fo* could also be defined as the community's ability to rebuild itself to the level of functionality it had before a disturbance (*Marzorati et al., 2008*).

To further assess similarity between the datasets produced by T-RFLP and Illumina sequencing, the T-RFs were rarefied without replacement to the lowest peak height, and OTUs to the lowest number of reads (Table S1), before Bray–Curtis inter-sample dissimilarity matrices were generated. These matrices were then subjected to principal co-ordinates analysis (PCoA), and a PERMANOVA (a one-way nonparametric permutational multivariate analysis of variance, (*Anderson, 2001*), with 999 permutations was used to test the effects of year, month, nest and island on the matrices. To ensure that the results for the bacterial OTUs were robust (*Morton et al., 2017*), PCoAs were also performed on Bray–Curtis matrices constructed from log transformed data, and on Morisita and Jaccard distance matrices. The uniformity of the T-RFLP and the OTU matrices was tested with Mantel's tests (method 'pearson', significance based on 999 permutations). Rarefying and the analyses described above were carried out in R with package vegan (*Oksanen et al., 2011*). Finally, the relative abundance patterns of the (non-rarefied) T-RFs and the OTUs were compared, with the expectation that both techniques would show similar proportions of abundances. The T-RFs (enzymes analysed separately), and the equivalent number of OTUs were ranked from the highest to the lowest (T-RFs according to their height, and the OTUs according to their read abundance), and the correlations (Pearson's *rho*) tested by $t$-test, were performed in JMP v.11 (SAS Institute Inc., Cary, NC, USA).

## Virtual T-RFs

To match the Illumina OTUs and their identities with the experimental T-RFs, a virtual restriction was performed on the 200 most abundant OTUs of each bacteria and fungi. These subsets were chosen based on the outcome of the correlation test, and the number of sequences they covered. The OTU sequences were first aligned (ClustalX2, *Larkin et al., 2007*) to confirm the coverage of the entire targeted 16s region (bacteria) or the ITS2 region (fungi) and the starting points of the primers were identified. Virtual T-RFs were generated with the Webcutter on-line tool (http://rna.lundberg.gu.se/cutter2/), for the two enzymes separately. The experimental T-RFs that were generated were matched with virtual T-RFs in Excel, and taxonomic identities were derived from the classification of the Illumina MiSeq sequences. A taxa was considered as being represented by the experimental T-RFs only if restriction sites for both enzymes were detected on the Illumina OTU sequences.

## RESULTS

The T-RFLP-analysis generated 129 (HaeIII) and 120 (MspI) unique bacterial T-RFs ($n = 37$), and 102 (HaeIII) and 124 (MspI) fungal T-RFs ($n = 36$), respectively (Table 1). The Illumina MiSeq sequencing yielded in total (i.e., for all samples) 1,896,920 bacterial reads (16S sequences), and 3,274,825 fungal reads (ITS2 sequences), of high quality ($n = 38$ for both bacteria and fungi) (Table S1 and SI_Bioinformatics). These clustered into 4,699 unique bacterial and 2,315 unique fungal OTUs (Table 1). The Good's estimate of coverage for the T-RF data suggested that 97.9% of the bacterial, and 93% of the fungal diversity had been captured by the method, whereas the coverage for the OTUs was 99.9% and 99.4%, respectively (Table 1). The species accumulation curves (Fig. 1) showed a similar pattern, with the curves for the T-RFs being generally less flattened towards the end of their curves, particularly so in the fungal data. Only the bacterial OTU communities reached asymptote, suggesting high sampling efficiency (Fig. 1). The mean Shannon-Wiener diversity index for the T-RFs was lower than for the OTUs for both the bacteria (3.35 vs. 5.69) and fungi (2.71 vs. 3.48) (Table 1). The Pareto-Lorenz curves suggested a high *Fo*, indicating a high functional organization, for both the bacterial and fungal communities, regardless of the method used (Fig. 2). In the bacterial data, 20% of the T-RFs and OTUs accounted for 85% and 87% of the species abundance, respectively (Fig. 2A). The corresponding values for the fungal T-RFs and OTUs projected 73% and 95%, respectively (Fig. 2B).

The PCoAs indicated clustering of bacterial communities within islands and nests, in both the T-RF and OTU data, but the clustering was more pronounced in the OTU data (Figs. 3A and 3B). The horseshoe pattern in the bacterial NGS data remained in the PCoA analysis when it was repeated with log-transformed data, and also when the analysis was performed on alternative distance matrices (Figs. S2A–S2E), suggesting a true pattern rather than an artefact. A clustering by month (Fig. S3) was indicated for the T-RFs, which seemed mainly driven by the values of September. No such indication was seen for the OTUs. The PERMANOVA confirmed the effect of month for the bacterial T-RFs. The PERMANOVA suggested compliance between the two methods, showing significant effects of island and nest for both the T-RF and OTU data (Table 2). The effect of year
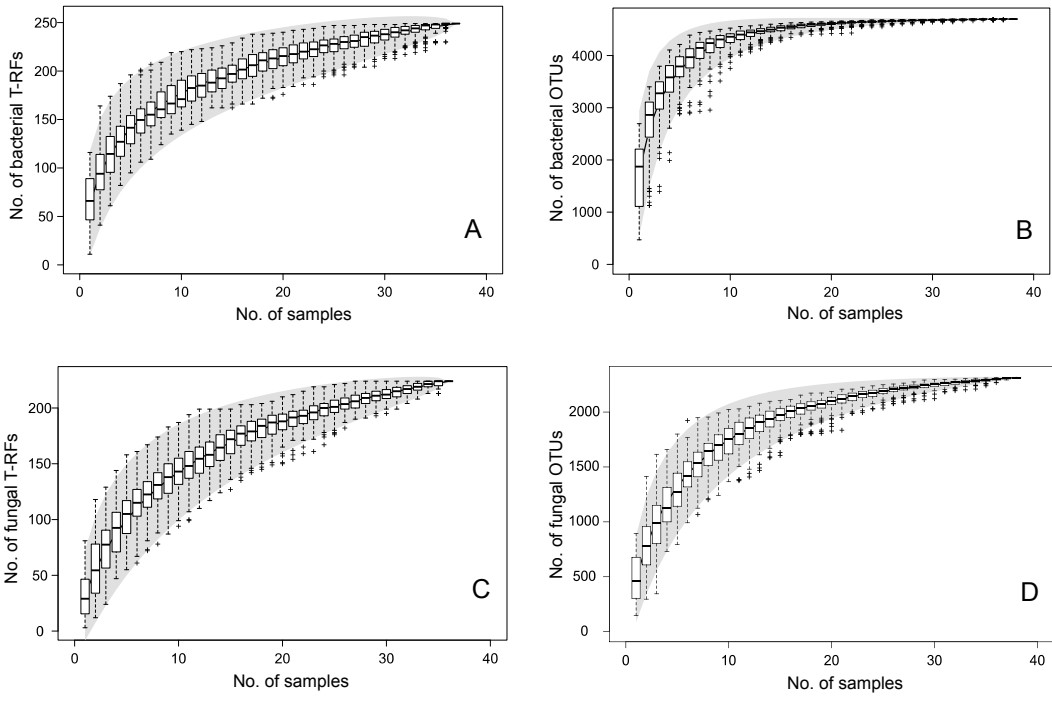

**Figure 1 Species accumulation curves for bacterial T-RFs (A) and OTUs (B), and fungal T-RFs (C) and OTUs (D).** The line represents the actual sampling, the grey area depicts the standard deviation and the box plot shows the species richness based on linear intrapolation of random permutations.

**Table 1 The number of T-RFs and OTUs obtained, Shannon-Wiener diversity index (*H′*) and Good's estimate of sampling coverage.**

|  | T-RFs | *SD* | T-RFs HaeIII | T-RFs MspI | OTUs | *SD* |
|---|---|---|---|---|---|---|
|  |  |  | **Bacteria** |  |  |  |
| No. of | 249 |  | 129 | 120 | 4,699 |  |
| Mean no. of (per sample) | 64 | 27 |  |  | 1,743 | 599 |
| Mean value of Shannon-Wiener *H′* | 3.35 | 0.43 |  |  | 5.69 | 0.67 |
| Good's estimate | 98.0% |  | 90.6% | 91.1% | 99.9% |  |
|  |  |  | **Fungi** |  |  |  |
| No. of | 226 |  | 102 | 124 | 2,315 |  |
| Mean no. of (per sample) | 32 | 20 |  |  | 482 | 14 |
| Mean value of Shannon-Wiener *H′* | 2.71 | 0.76 |  |  | 3.48 | 0.78 |
| Good's estimate | 95.0% |  | 74.% | 93.0% | 99.4% |  |

**Notes.**
T-RFs, $n = 37$ (bacteria) and $n = 36$ (fungi); OTUs, $n = 38$ for both bacteria and fungi.

was not significant, neither for the T-RFs nor the OTUs (Table 2). The PCoAs from the fungal T-RF and OTU data indicated clustering across both island and nest (Figs. 3C and 3D), with the fungal OTU data showing a much more pronounced clustering than the T-RF data (Figs. 3C and 3D). The PERMANOVA showed significant effects of island and nest for both T-RFs and OTUs in the fungal data, suggesting full compliance between the two methods for all the tested effects (Table 2). The Mantel tests showed moderate to

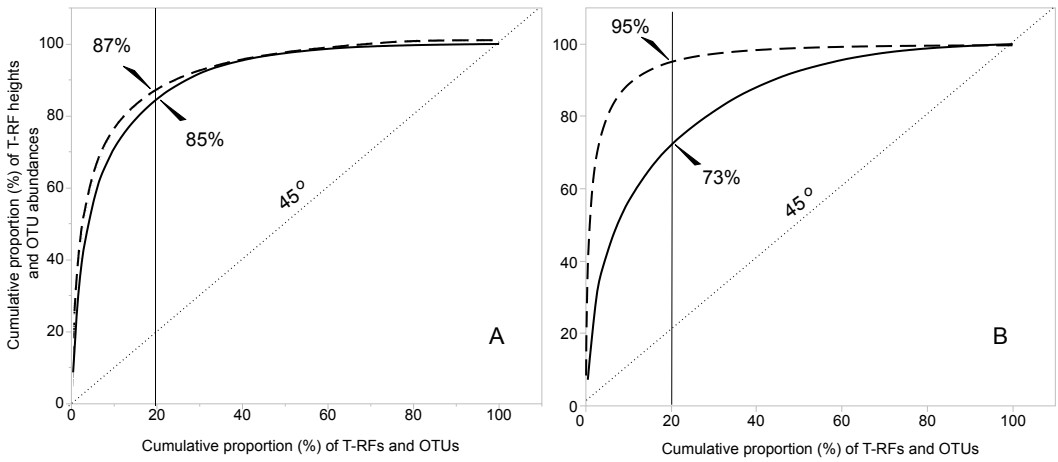

**Figure 2** **Functional organization (*Fo*) according to the Pareto-Lorenz curves of the bacterial (A) and fungal (B) T-RFs and OTUs. The continuous lines represent the T-RFs and the dashed lines the OTUs.** The value of *Fo* equals the value projected at the *y*-axis where the critical 20% at the *x*-axis (thin continuous line) intercepts with the Pareto-Lorenz curves. The dotted line shows the 45° slope, representing perfect evenness. A *Fo* value of 25% represents a community of high evenness with no distinct structure in terms of species dominance. A community at the *Fo* value of 45% is more functionally organized due to its evenness, and a value of 80% for the *Fo* stands for a highly specialized community, dominated by a low number of species on which the functional stability depends.

good correspondence between the T-RF and the OTU data, being significantly correlated (Pearson's *rho* = 0.743, $P \leq 0.001$, and 0.574, $P \leq 0.001$) for the bacterial and the fungal data sets, respectively.

The relative abundances of the T-RFs, and the equivalent number of OTUs, ranked highest to lowest (encompassing 52% (HaeIII) and 50% (MspI) of the bacterial, and 78% and 81% of the fungal OTU sequences), were highly and significantly correlated (Pearson's *rho* = 0.949 (HaeIII), and 0.849 (MspI), respectively) for the bacterial data, and 0.953 and 0.904, respectively for the fungal data; $P \leq 0.001$ in all cases; Figs. 4A–4D. Of all the bacterial OTUs, 89% were assigned identities at the phylum, and 53% at the family level (Table 3), whereas the corresponding values for the fungal OTUs were 85%, and 39%, respectively (Table 3, Figs. S4A–S4B).

The 200 most abundant bacterial OTUs, on which the virtual restriction was performed, comprised approximately 60% of all bacterial OTU sequences. Taxonomic information (based on the taxonomic identification of the Illumina OTUs) was available for 66% of the 200 bacterial OTUs down to the level of family. The enzyme HaeIII successfully cut nearly all (99%) of the most abundant 200 OTUs, whereas MspI cut 86%. Matching virtual T-RF with the T-RFs generated from soil samples returned 93 taxonomic items, with restriction patterns from both enzymes (Table S2). Seventeen different restriction patterns were found, three of which were shared between several phyla, whereas 14 patterns were unique at the order level, and six of these at the family level (Table S2). No taxonomic information was available for three of the restriction patterns below the class level, and one pattern returned phylum only. Four of the six identified families (Burkholderiaceae, Caulobacteraceae,
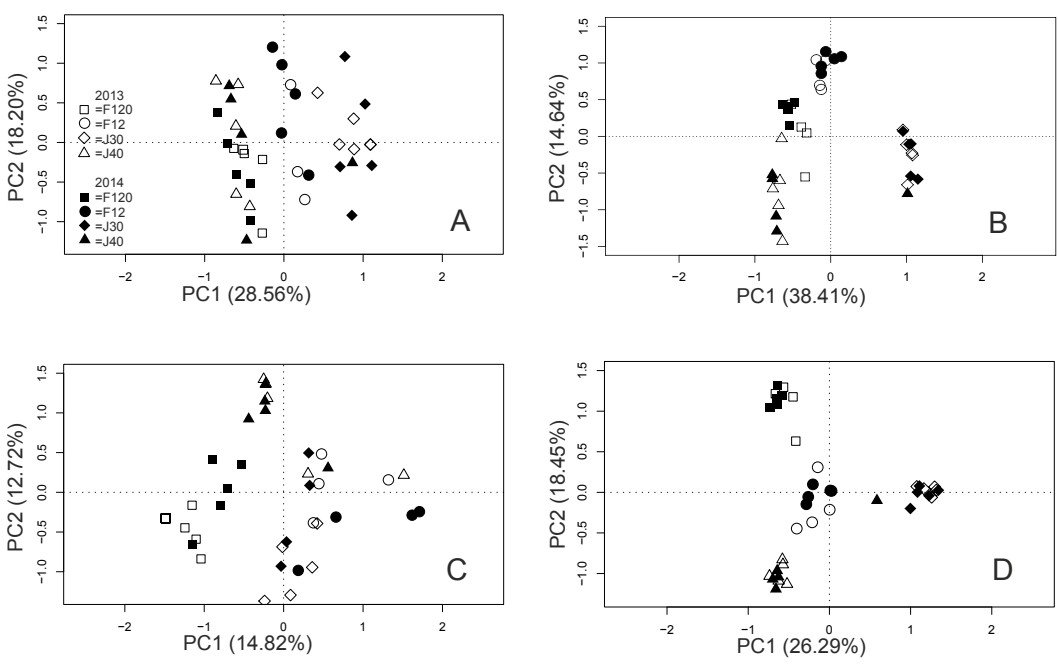

**Figure 3** **PCoA of bacterial T-RFs (A) and OTUs (B), and fungal T-RFs (C) and OTUs (D).** Year is indicated by colour and nest by shape of symbols as indicated in the graph.

**Table 2** **PERMANOVA test of the temporal (year and month) and spatial (island and nest) effects on the Bray-Curtis distances of the bacterial and fungal T-RF and OTU data.**

| Test effect | T-RFs | | | | OTUs | | | |
|---|---|---|---|---|---|---|---|---|
| | $F$ | $df$ | $R^2$ | $p$ | $F$ | $df$ | $R^2$ | $p$ |
| | | | | **Bacteria** | | | | |
| Year | 1.08 | 36 | 0.030 | 0.336 | 0.49 | 37 | 0.014 | 0.873 |
| Island | 2.50 | 36 | 0.067 | 0.021 | 6.84 | 37 | 0.160 | 0.001 |
| Nest | 3.58 | 36 | 0.093 | 0.001 | 4.20 | 37 | 0.104 | 0.005 |
| Month | 2.55 | 36 | 0.068 | 0.020 | 1.32 | 37 | 0.035 | 0.210 |
| | | | | **Fungi** | | | | |
| Year | 1.12 | 35 | 0.032 | 0.284 | 0.85 | 37 | 0.023 | 0.555 |
| Island | 2.99 | 35 | 0.081 | 0.001 | 7.46 | 37 | 0.172 | 0.001 |
| Nest | 4.65 | 35 | 0.120 | 0.001 | 7.31 | 37 | 0.169 | 0.001 |
| Month | 0.94 | 35 | 0.027 | 0.558 | 1.01 | 37 | 0.027 | 0.378 |

Nocardioidaceae and Streptomycetaceae) belonged to the bacterial families ranked as the most abundant according to the Illumina MiSeq data (Table 3).

The corresponding fungal data, based on the 200 most abundant fungal OTUs, on which the virtual restriction was performed, comprised approximately 87% of all fungal OTU sequences Taxonomic information down to the family level was available for 50% of the fungal OTUs. A virtual HaeIII restriction site was found in 97%, and an MspI restriction site in 93% of the fungal OTUs. Matching the virtual T-RFs with the T-RFs generated from

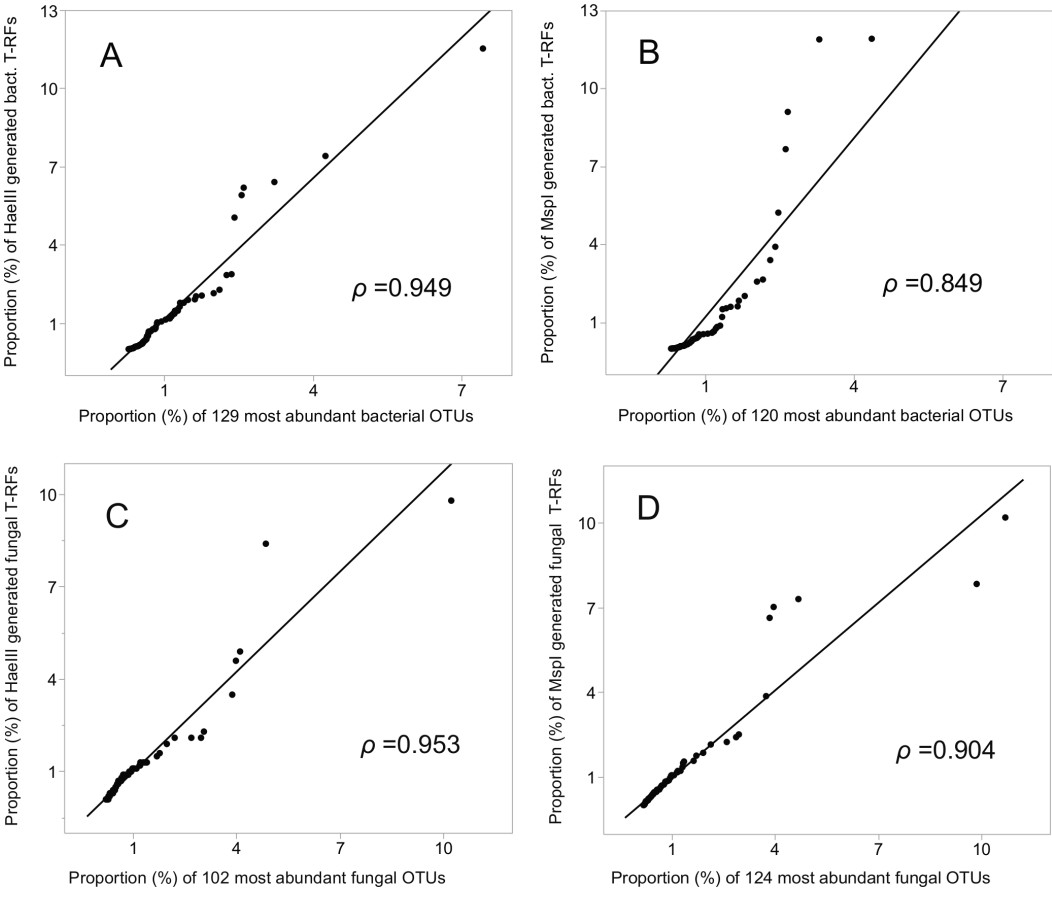

**Figure 4** **Correlation (Pearson *rho*) of T-RFs with the equivalent number of the most abundant OTUs, both ranked highest to lowest.** (A) HaeIII, (B) MspI generated bacterial T-RFs, (C) and (D) HaeIII and MspI generated fungal T-RFs respectively.

nest mound soil samples returned 41 taxonomic items, with restriction patterns from both enzymes. These taxonomic items represent 14 distinct restriction patterns, all of which were unique at the level of phylum (Table S3). Nine unique patterns were identified at the order level, and 8 patterns were further identified to the family level. Four patterns were identified to phylum only. Of the identified families, the Tremellales_incertae_sedis and Venturiaceae were among the most abundant ones according to the Illumina MiSeq data (Table 3).

## DISCUSSION

In this study, T-RFLP fingerprinting and Illumina sequencing were used to analyze the bacterial and fungal communities in the nest mounds of the ant *F. exsecta*. The results show that the combination of the two approaches is well suited to study microbial communities of high complexity, and that they provide similar, yet complementary information. With both methods, the microbial communities showed high and similar functional organization, and similar spatiotemporal patterns across both nest mounds and sampling occasions.

**Table 3 The number of the Illumina MiSeq OTUs assigned to the different taxonomic levels.** The five most abundant bacterial phyla and families, together with the four fungal phyla present in the data and the five most abundant fungal families.

| Taxonomic level | Nbr of OTUs assigned | |
|---|---|---|
| | **Bacteria** | **Fungi** |
| Kingdom | 4,699 | 2,315 |
| Phylum | 4,163 | 1,963 |
| | The five most abundant bacterial phyla: | The four fungal phyla present in data: |
| | Proteobacteria | Ascomycota |
| | Actinobacteria | Basidiomycota |
| | Bacteroidetes | Mucoromycota |
| | Acidobacteria | Chytridiomycota |
| | TM7 candidate phylum | |
| Class | 3,836 | 1,326 |
| Order | 3,159 | 1,105 |
| Family | 2,486 | 908 |
| | The five most abundant bacterial families: | The five most abundant fungal families: |
| | Streptomycetaceae | Herpotrichiellaceae |
| | Acetobacteraceae | Sporidiobolales_Incertae_sedis |
| | Nocardioidaceae | Venturiaceae |
| | Caulobacteraceae | Tremellales_Incertae_sedis |
| | Burkholderiaceae | Mortierellaceae |
| Genus | 2,031 | 725 |

Comparisons between fungi and bacteria also revealed that bacterial communities were more comprehensively sampled than fungal communities.

The number of T-RFs obtained in this study were similar in magnitude to other comparable studies of soil microbial communities in polar, pasture, meadow, agricultural and forest soils, both for bacteria (*Blackwood et al., 2003*; *Frey et al., 2009*; *Anderson et al., 2011*; *Van Dorst et al., 2014*), and fungi (*Schwarzenbach, Enkerli & Widmer, 2007*; *Boots et al., 2012*; *Van Dorst et al., 2014*). Similarly, the number of microbial sequences, the sum of refined OTUs, and the resultant unique OTUs identified in this study, were in good correspondence with recent NGS studies of soil bacteria (*Lazzaro, Hilfiker & Zeyer, 2015*; *Supramaniam et al., 2016*) and fungi (*Meiser, Bálint & Schmitt, 2014*). In general, the number of defined, unique OTUs was much higher than the number of respective T-RFs (approximately 36 times the bacterial, and 19 times the fungal T-RFs). Nevertheless, when ranked per relative abundance, the high Pearson correlation suggest similar proportions of the most abundant T-RFs and OTUs, for both bacteria and fungi. This does not automatically mean that the most abundant T-RFs correspond with the most abundant OTUs or vice versa, but it indicates that the most abundant taxa were obtained by both techniques, in similar relative proportion.

Comparisons between the sampling coverage, i.e., Good's estimates, and the species accumulation curves of fungi and bacteria, revealed that bacterial communities were more comprehensively sampled than fungal communities. Notably, the asymptote was not reached for neither fungi nor bacteria using T-RFLP, suggesting undersampling by this technique (*Zhou et al., 2008*). Only a small fraction of microbial diversity is sampled with this (and similar) methods (*Jankowski, Schindler & Horner-Devine, 2014*), resulting in low detection sensitivity and flatter species accumulation curves. Comparatively, the asymptote was reached for bacteria, and the curve better saturated for fungi using NGS, highlighting the power of this method to detect rarer taxa. Consequently, the diversity estimates were also lower for fungi than for bacteria. Technical factors likely to contribute to the observed differences in sampling success include the length of sequenced fragments, the choice of genes sequenced, and primer specificity. The technical requirement of using different primer pairs for both methods, resulted in fragments of differing length. The bacterial Illumina sequences were ~500 bp (*Timonen et al., 2017*), and the T-RFLP sequences ~1,200 bp (*Muehling et al., 2016*) long, respectively, whereas the fungal target sequence was 280–310 bp long (*Ihrmark et al., 2012*) for both methods. Thus the longer fragment can be expected to provide more phylogenetic information, leading to a finer tuned OTU clustering. For the T-RFLP differentiation, the longer fragment could provide an increased number of enzyme cutting sites, which could partly explain the higher yield of the bacterial T-RFs compared to the fungal T-RFs. In addition, the bacterial 16S gene is highly conserved, and successfully targets most bacterial phyla (*Větrovský et al., 2013*). Developing fungal primers with similar attributes has proven more challenging (*Martin & Rygiewicz, 2005*; *Asemaninejad et al., 2016*), as fungal markers fulfilling the criteria of being of suitable length and specificity, yet not discriminating against any fungal taxa are not available (*Lindahl et al., 2013*). This is further compounded by the fact that a substantial proportion of the fungal ITS sequences in the International Nucleotide Sequence Database are incomplete, with fragmentary regions in the ITS1 and ITS2 subsections (*Nilsson et al., 2009*). From a biological standpoint, it is nevertheless possible that the fungal communities are less diverse, owing to abiotic factors (*De Vries & Shade, 2013*; *Wagg et al., 2014*).

Illumina sequencing and T-RFLP fingerprinting captured the same patterns of functional organization in the nest mounds of *F. exsecta*. The Pareto-Lorenz curves suggested a high functional organization of the microbial communities inside the nest mounds. This suggests that the number of dominant species within the community is relatively low (*Wittebolle et al., 2009*), given that a high functional organization signals numerical dominance by a few species (*Mertens, Boon & Verstraete, 2005*; *Marzorati et al., 2008*; *Wittebolle et al., 2008*; *Gonzalez-Gil & Holliger, 2011*). This agrees with our finding that 50% of the fungal sequences, and 60% of the bacterial sequences were found among the 200 most abundant OTUs. Such communities are often specialized, but potentially susceptible to disturbance, requiring a long recovery time for regaining their functionality. Conversely, a community of high species evenness is expected to cope better with environmental disturbances (*Marzorati et al., 2008*). The high level of functional organization suggests that the microbial communities in the nest mounds may be specialized and strongly affected by the habitat with which they are associated (*Edwards et al., 2011*; *Boots & Clipson, 2013*;

*Cai et al., 2014*). Further study to confirm an existent influence of the *F. exsecta* nest environment on the bacterial and fungal communities, compared with the effect of a reference substrate, would be required.

Both techniques also captured similar community patterns in bacteria and fungi. The results suggest that microbial communities were primarily structured at the level of nest mounds, with a more distinct clustering in the OTU data. Structure at the level of island was also found, but this effect may be compromised by the low sample size, and a more extensive study with a larger number of sampling sites would be required to confirm this effect. In this study, the higher sequencing depth (i.e., higher number of OTUs generated) likely contributed to the slightly lower correspondence between NGS and T-RFLP matrices, compared to that found in e.g., *Pilloni et al. (2012)* and *De La Fuente et al. (2014)*. A higher sequencing depth is likely to reveal rarer taxa, which would not have been defined in earlier studies, and are inaccessible to methods such as T-RFLP.

Although even greater taxonomic details for the T-RFs, and a better agreement between the two methods could be achieved by the use of additional enzymes (*Dunbar, Ticknor & Kuske, 2001*; *Edwards & Turco, 2005*), the overall taxonomic information obtained was comparable to those reported for extensive T-RFLP studies involving clone libraries (*Axelrood et al., 2002*; *Youssef & Elshahed, 2009*; *Lin et al., 2011*). The correlation between the relative abundance of T-RFs, and the equivalent number of the most abundant OTUs was significant both for bacteria and fungi, which corroborates the congruence between the methods. The outcome of the correlation served as a workable guideline for the number of OTUs (200) that were forwarded to the *in silico* T-RF cuttings. For both bacteria and fungi, the virtual T-RFLP generated restriction patterns from the majority of this subset of OTU sequences obtained by the Illumina sequencing. The bacterial OTUs were assigned to 13 different phyla, with roughly half (52%) being assigned to the phyla of Proteobacteria and Actinobacteria, both of which were also captured by the matching of T-RFs. At lower taxonomic levels, T-RFs and the virtual restriction cutting patterns corresponded to the taxonomic families of Burkholderiaceae, Caulobacteraceae, Nocardioidaceae and Streptomycetaceae. In the fungal data, the Illumina sequencing captured four phyla, whereas, based on the virtual restriction, the T-RFLP only captured two phyla, the Ascomycota and Basidiomycota. At lower taxonomic levels, the virtual restriction captured the families Tremellales incertae sedis, and Venturiaceae, both of which were the most abundant fungal families captured by the Illumina sequencing.

The bacterial and fungal taxa captured by our analysis are abundant in soil environments (*Acosta-Martínez et al., 2008*; *Buée et al., 2009*; *Anderson et al., 2011*; *Voglmayr et al., 2011*; *Lanzén et al., 2015*; *DeAngelis et al., 2015*). Furthermore, the bacterial genera belonging to Burkholderiaceae have also been shown to be associated with ant genera such as, *Camponotus* (*He et al., 2011*), *Cephalotes* (*Russell et al., 2009*; *Kautz et al., 2013*), and *Tetraponera* (*Van Borm et al., 2002*). Several genera representing both Proteobacteria and Actinobacteria, and sequences matching *Burkholderia* were also present in the transcriptomic data of *F. exsecta* (*Johansson et al., 2013*). The two fungal families that were captured by both T-RFLP and Illumina (Tremellales incertae sedis and Venturiaceae) are not currently defined in the literature as having any association with ants. However, the

literature on fungi associated with ants is focused solely on the fungus-farming ants and the restricted number of fungal species they cultivate, or the few insect pathogenic fungi studied in connection with ants (*Hughes et al., 2004*).

## CONCLUSIONS

Thorough assessment of complex microbial communities is challenging, and the use of multiple methods is often required in order to achieve a comprehensive outcome. This study describes certain steps at which technical choices during the process have substantial impact on the outcome, pinpointing some plausible reasons for non-conformity between the two methods. Overall, this study shows that both T-RFLP and Illumina sequencing are suitable for analysis of the fungal and bacterial communities in nest mounds of ants, but when the two techniques are combined, it can provide an even more robust dataset. Furthermore, this study validates the use of both techniques when addressing the topic of microbial communities in complex environments such as ant nest mounds, both independently or to support findings within comparable studies. In particular, when comprehensive longitudinal studies based on T-RFLP exist, supplementing the fingerprinting data with NGS is a practical solution. Although refining the results gained by the two methods into a reliable synthesis requires testing of overall compliance, T-RFLP can be trusted to reveal the same general community patterns as NGS (Illumina MiSeq), and is a good option if resources for NGS are limited.

## ACKNOWLEDGEMENTS

We thank our technician Heini Ali-Kovero for her valuable input during the laboratory assays, Minna-Maarit Kytöviita for her kind assistance in identifying the soil type and typical plants of the sampling site, Claire Morandin for her valuable help with the bioinformatics and Hanna Sinkko for all her statistical support.

### Funding

This work was funded by the Academy of Finland (grant numbers 252411 and 284666). The funders had no role in study design, data collection and analysis, decision to publish, or preparation of the manuscript.

### Grant Disclosures

The following grant information was disclosed by the authors:
Academy of Finland: 252411, 284666.

### Competing Interests

The authors declare there are no competing interests.

## Author Contributions

- Stafva Lindström conceived and designed the experiments, performed the experiments, analyzed the data, prepared figures and/or tables, authored or reviewed drafts of the paper, approved the final draft.
- Owen Rowe and Sari Timonen conceived and designed the experiments, authored or reviewed drafts of the paper, approved the final draft.
- Liselotte Sundström conceived and designed the experiments, contributed reagents/materials/analysis tools, authored or reviewed drafts of the paper, approved the final draft.
- Helena Johansson conceived and designed the experiments, performed the experiments, authored or reviewed drafts of the paper, approved the final draft.

## Field Study Permissions

The following information was supplied relating to field study approvals (i.e., approving body and any reference numbers):

Field experiments were approved by Helsinki University, Tvärminne Zoological station.

## DNA Deposition

The following information was supplied regarding the deposition of DNA sequences:

The sequences are deposited at NCBI Sequence Read Archive, project number PRJNA399258.

## Data Availability

The seq data and the t-rflp data are uploaded as Supplemental Files.

## Supplemental Information

Supplemental information for this article can be found online at http://dx.doi.org/10.7717/peerj.5289#supplemental-information.

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
