# Peer review of "Trends in bacterial and fungal communities in ant nests observed with Terminal-Restriction Fragment Length Polymorphism (T-RFLP) and Next Generation Sequencing (NGS) techniques—validity and compatibility in ecological studies"

_PeerJ, doi:10.7717/peerj.5289_

## Round 0.1 · original submission · Major Revisions

Please pay close attention to the comments from Reviewer 1.

Reviewer 1 ·

Basic reporting

Some of the methodological details were unclear to me, but most of the text was well-written.
The captions for the tables and figures need to be more comprehensive. I was confused by some of them and they are not self-contained.
For the fungi, the primers were the same for T-RFLP and MiSeq (fITS7 and ITS4). However, the bacterial primers differ (T-RFLP: 27F and 1387R; MiSeq: 27F, pD). I would like an explanation of why the bacterial primers differed for the different methods and outline how this could impact results in the Discussion.
How TRFs were aligned to OTUs is not clear to me. How were the sequences aligned with the virtual TRFs? Which program? How similar were sequences from MiSeq considered to be the same as the virtual taxa?
I am confused about the functional organisation analysis and why it is necessary.
Bioinformatics: I need a bit more information here. There is no information on the quality filtering and the number of reads. This is useful information for people who may want to use these methods in the future so they can get an idea of the necessary sampling depth. Also, what length were the sequences cut to before clustering? Was any denoising done?

Experimental design

No comment

Validity of the findings

I would argue that the curves do not show similar patterns in the bacterial communities for the different methods and that MiSeq has done a better job of capturing this. Note that the curves do not reach asymptote when using TRFLP for either bacteria or fungi – this should be discussed in the Discussion because it suggests undersampling.
“Regardless of the molecular method, bacterial communities were more comprehensively sampled than fungal communities, judging by how rapidly the asymptote was reached by the bacterial vs. the fungal curves (Fig 2).” – This is not true – asymptote was only reached for the bacteria in MiSeq
Some more discussion on how this approach compares to the traditional cloning library would also be worthwhile.

Additional comments

Lindström et al provide a novel methodological insight into comparing a traditional DNA fingerprinting technique, T-RFLP, with next generation sequencing, Illumina MiSeq, in bacterial and fungal communities from ant nests. They found that there was reasonably strong congruence in results when using both methods, but more taxa were able to be delineated from the MiSeq, which they note as unsurprising.

This is a topic of interest and the experimental design and methods are generally well-written. I found the first part of the manuscript very well written and clear, but there needs to be more detail in some parts of the methods and results for me to be able to follow how things were done.

One aspect that is mentioned in the introduction but provides clear support for the approach is the possible benefit of combining NGS with T-RFLP in longitudinal studies. For example, the first measurement could use both methods to get comprehensive taxonomic information and relate NGS OTUs to TRF virtual taxa, then in subsequent sampling we may be able to rely on just TRFLP. It could potentially overcome some of the criticisms involved in fingerprinting alone because we would still have taxonomic information, enabling us to discuss ecology. This could substantially reduce costs and I think this is a key strength that supports the novelty of the study.

I found most of the data analyses to be well-chosen but it is unclear how the virtual TRFs were assigned to taxa, which is clearly a very important part of the study. The functional organisation measure is unclear to me and I am not sure what the information it provides other than information about dominance, which seems to be easy enough to assess by just looking at read counts.

The captions for the tables and figures need to be more comprehensive. I was confused by some of them and they are not self-contained. The readers need to be reminded how many samples are being analysed and also that OTUs are from the MiSeq.

I suggest depositing the raw data into an online database, such as dryad, rather than providing it as supplementary material.

Introduction
L52: It should be noted that quantitative information from these fingerprinting methods is not always considered appropriate, requiring rigorous standardisation techniques that are not always used (Green SJ, Leigh MB, Neufeld JD (2009) Denaturing Gradient Gel Electrophoresis (DGGE) for Microbial Community Analysis. In: Handbook of Hydrocarbon and Lipid Microbiology (ed Timmis KN), pp. 4137–4158. Springer, Heidelberg, Germany). This is why many authors choose to use presence-absence. The same concerns have been raised about NGS data (Amend AS, Seifert KA, Bruns TD (2010) Quantifying microbial communities with 454 pyrosequencing: does read abundance count? Molecular Ecology, 19, 5555–5565)

L60: NGS acronym needs to be defined.
L80: “Similarly to microbes…”. I suggest removing this part of the sentence because microbes are much more ubiquitous than ants
This paragraph could be linked better with the molecular method information in the first paragraph. Perhaps bringing the microbial relationships with ants into the first sentence as a link would make this link clearer. After reading the rest of the manuscript, I think this paragraph could potentially be cut down, since this information is not really relevant for the methodological aspect of the study.

Methods
L144: “lusher patches of vegetation” is vague. I suggest giving a broad overview of common plants in the area and/or state the biome. I am also interested in the soil type, given that the ants nest in soils.
I suggest a slight restructuring of the Methods. Combine subheadings “DNA extraction and PCR amplification for T-RFLP” and “Analysis of T-RFLP data” into “T-RFLP Analysis” subheading. I suggest the same for the MiSeq methods. Also, put the MiSeq part first, then the T-RFLP information, so that the virtual T-RFLP section is encompassed in the larger T-RFLP section.
For the fungi, the primers were the same for T-RFLP and MiSeq (fITS7 and ITS4). However, the bacterial primers differ (T-RFLP: 27F and 1387R; MiSeq: 27F, pD). I would like an explanation of why the bacterial primers differed for the different methods and outline how this could impact results in the Discussion.
Bioinformatics: I need a bit more information here. There is no information on the quality filtering and the number of reads. This is useful information for people who may want to use these methods in the future so they can get an idea of the necessary sampling depth. Also, what length were the sequences cut to before clustering? Was any denoising done?
L193: I am inferring that the most common 200 OTUs were chosen because these are the ones that would be expected to be captured in the T-RFLP analysis. I’m interested why this number was chosen and if other numbers were tried – different iterations of this could certainly strengthen the validity of this approach.
L197: This part is not clear to me. How were the sequences aligned with the virtual TRFs? Which program? How similar were sequences from MiSeq considered to be the same as the virtual taxa?
L204: Please provide version numbers for the packages used and provide a citation for R. It is also helpful to provide the function names. E.g. I cannot find a function in vegan for the Good’s estimate.
L212: What does this functional organisation analysis tell us? Does it tell us about functional groups?
L228: Which dissimilarity measure was used in the PCoA?
L232: At what point was rarefying done? Prior to all of the statistical analyses, and to what depth?
Was there any point where the full OTU information from the MiSeq were analysed? It is not clear from the statistical methods.
L235: spelling: “abundancies” change to “abundances”

Results
L247: I would argue that the curves do not show similar patterns in the bacterial communities for the different methods and that MiSeq has done a better job of capturing this. Note that the curves do not reach asymptote when using TRFLP for either bacteria or fungi – this should be discussed in the Discussion because it suggests undersampling.
“Regardless of the molecular method, bacterial communities were more comprehensively sampled than fungal communities, judging by how rapidly the asymptote was reached by the bacterial vs. the fungal curves (Fig 2).” – This is not true – asymptote was only reached for the bacteria in MiSeq

L271: Suggest adding this to the end of the previous paragraph and reframing: “The Mantel tests showed good correspondence between the T-RF and the OTU data, being significantly correlated for the bacterial (R = 0.743, P≤0.001) and fungal (R= 0.574, P≤0.001) data sets.” – It is worth noting that although the correlations are significant, they are not really very high.

Table 1: I am confused by the headings here. What do the “T-RFs HaeIII T-RFs MspI” columns relate to and how do they differ from the TRFs column overall? I think it also needs to be clearer here that the OTUs are from MiSeq. Please provide more detail and remind us how many samples were analysed.
Table 2: Why are the df different for the TRF analyses. Were some samples lost? It is very interesting that the same effects come out as significant, regardless of the method.

Fig 1: I find this map is not very helpful without an outline of the wider area.
Fig 2: These graphs are great. I find it very interesting that the species accumulation for fungi is similar for the two methods, but the MiSeq has clearly done better in the bacterial community in fewer samples. The curves do not reach asymptote when using TRFLP for either bacteria or fungi. This warrants mention.
Fig 3: I do not understand these graphs and need more interpretation in the text.
Fig 4: It is not clear how the monthly samples lay over this and it seems that the sites are not different in different years. There are also clearly artefacts in the ordination in 4B and D, with curves. I suggest trying to transform the data or try a different dissimilarity measure.

Discussion
The discussion can use some re-structuring and focus. The interesting information is all there but it is difficult to find. I would focus the first part of the discussion heavily on the aims of the study, i.e. the comparison of MiSeq and TRFLP and how they aligned and where they did not. Then after this, move on to the patterns that were shown to be similar in both methods (i.e. the PERMANOVAs and PCoA). Then I would move on to how these taxa relate to wider studies. At the moment this all seems a bit jumbled together.
Really, the key outcome is that T-RFLP can tell us about common species in the communities, and these are the ones that drive the multivariate analyses used. Indeed, often the rare species are removed from NGS data in order for us to see a clearer pattern (it is not clear if this was done for the current data).
Some more discussion on how this approach compares to the traditional cloning library would also be worthwhile.
L324: spelling “reactive” to “relative”
L334: For MiSeq, the length the sequences were cut in the bioinformatics is more important than the primer length, in terms of getting phylogenetic information.
The second paragraph of the discussion is comparing bacterial and fungal communities. I feel like this detracts from the aim of the paper and the TRF vs MiSeq comparison is lost in the detail here.
L353: ‘high functional organization signals numerical dominance by a few species” – this explanation of the metric should be moved to the Methods.
L355: “This agrees with our finding that 50% of the fungal sequences, and 60% of the bacterial sequences were found among the 200 most abundant OTUs.” – This statement makes me wonder the value of the functional organisation method, since this more simple assessment has provided the same answer. I had also expected the functional organisation analysis to provide information about functions of taxa in the communities, but it seems to only provide information about dominance.

Reviewer 2 ·

Basic reporting

Paper by Lindström et al. is well written, clearly structured and research questions well defined. If there is a weaknesses it is in the presenting results. Firstly, there is many figures and tables, could you combine some of them or even provide information in the text? For example, Figure 1 could be transferred in the supplementary material, also a small map of Finland could be added to the figure 1 to illustrate where study areas in Finland are located. Table 2 could be also added to supplementary material, however the key PERMANOVA results including F, P and df should be added in the text. Also, figure 4 would need ellipses to point the key results in the figure i.e. island and nest. Furthermore, figure 4 is missing x and y –axis labels, they should be added. Please, report P-values in italics.

Experimental design

Article meets PeerJ standards. Please specify why 200 most abundant OTU’s were used to match the Illumina OTU’s and their identities with the experimental T-RFs. Also, specify used 16S reverse primer for T-RELP and Illumine MiSeq, do they differ between the methods? Line 229 specify PERMANOVA is one-way nonparametric permutational multivariate analysis of variance and add citation to Anderson 2001. In line 232 do you mean rarefying?

Validity of the findings

Lindström et al. state that the bacterial and fungal communities in the F. exsecta mounds have not been studied previously, however community structure of fungal and bacterial communities in F. exsecta are not thoroughly discussed. In particular, Tremellales incertae sedis and Venturiaceae fungal families which are not currently defined as having any association with ants could be discussed further, this would increase novelty of the work and fills profoundly identified knowledge gap. Otherwise, advantages and disadvantages of the used methods are extensively discussed, also their validity and compatibility in ecological studies are well discussed. However, discussion could stated more clearly that the method used for studying microbial communities can have effect on the results i.e. for rare and perhaps endemic taxa. Therefore chosen method should be based on the aim of the ecological study. Also, cost and time of the methods could be discussed in more detailed.

Additional comments

Paper by Lindström et al. is well written, clearly structured and research questions well defined. The article clearly states and responses identified knowledge gap, however this could be improved with detailed discussion. It meets PeerJ standars, but still before mentioned small changes should be provided upon before acceptance.

---

## Round 0.2 · Minor Revisions

Please work through all the comments and clearly show how you have better explained the methods for your readers.

Reviewer 1 ·

Basic reporting

I appreciate the authors for the work they have in to account for my previous comments. I find the aims of this study very interesting and addresses very current issues surrounding molecular techniques. The changes that have been made have greatly clarified my previous concerns and have strengthened the manuscript.

Experimental design

The great contribution of this paper is to provide others with the ability to do TRFLP and have confidence that the general community patterns will be the same as the more expensive NGS MiSeq. Rewording the last sentence in the abstract and also emphasising this in the introduction and Discussion could strengthen the paper.

Validity of the findings

I still find the details of the bioinformatics sequencing too sparse for this study that is focussed on comparing molecular methods. These methods and decisions during sequence processing could greatly determine similarity of the results between TRFLP and MiSeq. I am not suggesting a reanalysis but more information on how it was done. E.g., what kind of quality control was done after the sequences were obtained? Were paired-end reads used and how? How long were the final sequences? How were chimeras detected and removed? How were OTUs defined (e.g., 97%)? What was the lowest number of reads (L217 indicates samples were rarefied to this)? What were the range in read numbers for different samples? Were samples with very low reads omitted?
This recent paper highlights some of the issues that should be covered: https://www.nature.com/articles/s41579-018-0029-9

Additional comments

Fig. 2. Thank you for clarifying the interpretation of these panels. In each graph, is the dotted line the results from the NGS and the solid line from TRFLP? Or vice-versa?

The addition of the explanation of the different primers used due to methodological considerations would be better placed in the Methods because this is where the reader is wondering why they are different.

---

## Round 0.3 · accepted · Accept

Thank you for your thorough responses to reviewer comments.

#